# Language Models as Hierarchy Encoders

**Yuan He**
University of Oxford
yuan.he@cs.ox.ac.uk

**Zhangdie Yuan**
University of Cambridge
zy317@cam.ac.uk

**Jiaoyan Chen**
The University of Manchester
jiaoyan.chen@mancheser.ac.uk

**Ian Horrocks**
University of Oxford
ian.horrocks@cs.ox.ac.uk

## Abstract

Interpreting hierarchical structures latent in language is a key limitation of current language models (LMs). While previous research has implicitly leveraged these hierarchies to enhance LMs, approaches for their explicit encoding are yet to be explored. To address this, we introduce a novel approach to re-train transformer encoder-based LMs as Hierarchy Transformer encoders (HITs), harnessing the expansive nature of hyperbolic space. Our method situates the output embedding space of pre-trained LMs within a Poincaré ball with a curvature that adapts to the embedding dimension, followed by training on hyperbolic clustering and centripetal losses. These losses are designed to effectively cluster related entities (input as texts) and organise them hierarchically. We evaluate HITs against pre-trained LMs, standard fine-tuned LMs, and several hyperbolic embedding baselines, focusing on their capabilities in simulating transitive inference, predicting subsumptions, and transferring knowledge across hierarchies. The results demonstrate that HITs consistently outperform all baselines in these tasks, underscoring the effectiveness and transferability of our re-trained hierarchy encoders.[1]

## 1 Introduction

In the field of Natural Language Processing (NLP) and related areas, the emergence of transformer-based language models (LMs) such as BERT (encoder-based) [1], GPT (decoder-based) [2], and the more recent large language models (LLMs) like GPT-4 [3] and Llama 2 [4], has marked a significant progression. Nonetheless, these models face a notable challenge in effectively encoding and interpreting hierarchical structures latent in language. This limitation has been highlighted by several studies, including those by [5] and [6], which employed prompt-based probes to reveal the limited hierarchical knowledge in pre-trained LMs, and the work by [7], which demonstrated these models' struggles with capturing the transitivity of hierarchical relationships.

Prior research has explored various methods to infuse hierarchical information into LM training. Common approaches include classification-based fine-tuning using sentence head embedding with a classification layer [8] or few-shot prompting with an answer mapping to classification labels [6]. To further pre-train, or re-train[2] LMs on a corpus constructed from hierarchical data, [9] converted

---

[1]See GitHub repository: `https://github.com/KRR-Oxford/HierarchyTransformers`; Datasets on Zenodo: `https://zenodo.org/doi/10.5281/zenodo.10511042` or the Huggingface Hub: `https://huggingface.co/Hierarchy-Transformers`; and HIT models also on the Huggingface Hub.

[2]In this work, the term *re-train* refers to train LMs on a new corpus without modifying its architecture; it is distinguished from standard fine-tuning that involves adding task-specific layers which lead to additional learnable parameters.

38th Conference on Neural Information Processing Systems (NeurIPS 2024).

structural representations into textual formats to align the masked language modeling objective. Others, like [10] and [11], have focused on extracting analogous and contrasting examples from hierarchical structures for a similarity-based contrastive learning objective. The aforementioned studies leveraged hierarchical information as implicit signals to augment LMs, yet no existing works specifically targeted the explicit encoding of hierarchies with LMs.

To bridge this gap, we introduce a novel approach to re-train transformer encoder-based LMs as **Hierarchy Transformer encoders** (HiTs). Inspired by the efficacy of hyperbolic geometry in representing hierarchical structures [12, 13], we propose the hyperbolic clustering and centripetal losses tailored for LM re-training. As illustrated in Figure 1, transformer encoder-based LMs typically use a $\tanh$ activation function in the last layer, which maps each embedding dimension to the range $[-1, 1]$. Consequently, the output embeddings of LMs are confined within a unit $d$-dimensional hypercube. Leveraging this characteristic, we utilise a Poincaré ball of radius $\sqrt{d}$, whose boundary circumscribes[3] the output embedding space of LMs. The metrics for distance and norm used in our hyperbolic losses are defined w.r.t. this specific manifold. After re-training, entities are not only clustered according to their relatedness but also hierarchically organised.

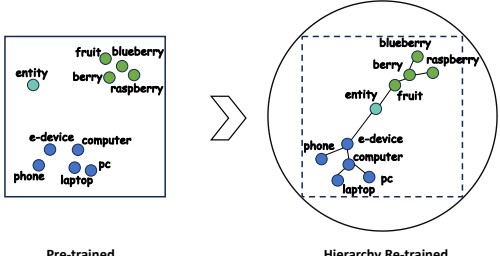

Pre-trained    Hierarchy Re-trained

Figure 1: Illustration of how hierarchies are explicitly encoded in HiTs. The square ($d$-dimensional hyper-cube) refers to the output embedding space of transformer encoder-based LMs whose final activation function is typically $\tanh$, and the circumscribed circle ($d$-dimensional hyper-sphere) refers to the Poincaré ball of radius $\sqrt{d}$. The distance and norm metrics involved in our hyperbolic losses are defined w.r.t. this manifold.

In evaluating HiTs, we compare their performance against pre-trained LMs, standard fine-tuned LMs, and previous hyperbolic embedding models in the Multi-hop Inference and Mixed-hop Prediction tasks. The Multi-hop Inference task, following the setting in [13], involves training models on all asserted (i.e., one-hop) subsumptions and assessing their ability to infer transitive (i.e., multi-hop) subsumptions. The Mixed-hop Prediction task is designed to mirror real-world scenarios, where models trained on incomplete hierarchy are applied to predict unknown subsumption relationships between arbitrary entity pairs. Additionally, we introduce a transfer learning setting, where models trained on one hierarchy are tested on another. Our experiments utilise datasets derived from WordNet [14] and SNOMED CT [15],[4] and transfer evaluation datasets from Schema.org [16], Food Ontology (FoodOn) [17], and Disease Ontology (DOID) [18]. The results show that HiTs significantly surpass all baselines in these tasks, demonstrating their robustness to generalise from asserted to inferred and unseen subsumptions, and a promising potential in hierarchy-based semantic search.

## 2    Preliminaries

### 2.1    Language Models

Transformer encoder-based LMs excel in providing fine-grained contextual word embeddings for enhanced language understanding. A key component of these models is the self-attention mechanism, which dynamically assigns importance to different segments of the input text, thereby capturing nuanced contextual semantics more effectively. Notable examples of such models include BERT [1] and RoBERTa [19], both of which utilise the *masked language modelling* objective during pre-training. This approach involves partially masking input sentences and prompting the model to predict the masked tokens, using the unmasked surrounding text as context. For acquiring sentence-level embeddings, these models can be augmented with an additional pooling layer, applied over the token embeddings [20, 21]. Pooling strategies such as mean, max, and sentence head pooling are employed, with their effectiveness varying across different applications. A contrastive learning objective is often applied for refining sentence-level semantics [21, 22]. Despite the rise of generative LLMs,

---

[3]We ignore the vertices of the hyper-cube, as they are on the boundary and thus are undefined in the open Poincaré ball.

[4]Results on WordNet are presented in Section 4, while results on SNOMED CT are presented in Appendix D.

transformer encoder-based LMs maintain their importance, offering versatility and efficiency in tasks like text classification and semantic search.

## 2.2 Hyperbolic Geometry

Hyperbolic geometry, a form of non-Euclidean geometry, is featured by its constant negative Gaussian curvature, a fundamental aspect that differentiates it from the flat, zero curvature of Euclidean geometry. In hyperbolic space, distances between points increase exponentially as one moves towards the boundary, making it inherently suitable for embedding hierarchical structures. This intuition aligns with the tree embedding theorem based on $\delta$-hyperbolicity, as discussed in [23] and [13].

Among the various models[5] of hyperbolic geometry that are isometric[6] to each other, the Poincaré ball is chosen for its capacity to contain the the output embedding space of LMs directly, as explained in the second last paragraph of Section 1. The $d$-dimensional Poincaré ball with a negative curvature $-c$ (where $c > 0$) is defined by the open ball $\mathbb{B}_c^d = \{\mathbf{x} \in \mathbb{R}^d : \|\mathbf{x}\|^2 < \frac{1}{c}\}$. The distance function in this model, dependent on the curvature value $c$, is given by:

$$d_c(\mathbf{u}, \mathbf{v}) = \frac{2}{\sqrt{c}} \tanh^{-1}(\sqrt{c}\|-\mathbf{u} \oplus_c \mathbf{v}\|) \tag{1}$$

In this equation, $\mathbf{u}, \mathbf{v} \in \mathbb{B}_c^d$, $\|\cdot\|$ denotes the Euclidean norm, and $\oplus_c$ denotes the Möbius addition [24] defined as:

$$\mathbf{u} \oplus_c \mathbf{v} = \frac{(1 + 2c\langle \mathbf{u}, \mathbf{v} \rangle + c\|\mathbf{v}\|^2)\mathbf{u} + (1 - c\|\mathbf{u}\|^2)\mathbf{v}}{1 + 2c\langle \mathbf{u}, \mathbf{v} \rangle + c^2\|\mathbf{u}\|^2\|\mathbf{v}\|^2} \tag{2}$$

Here, $\langle \cdot, \cdot \rangle$ denotes the inner product in Euclidean space. Note that for flat curvature $c = 0$, $\mathbb{B}_c^d$ will be $\mathbb{R}^d$ and $\oplus_c$ will be the Euclidean addition.

## 2.3 Hierarchy

We define a hierarchy as a directed acyclic graph $\mathcal{G}(\mathcal{V}, \mathcal{E})$ where $\mathcal{V}$ represents vertices that symbolise *entities*, and $\mathcal{E}$ represents edges that indicate the *direct* subsumption relationships asserted in the hierarchy. We can then derive *indirect* subsumption relationships $\mathcal{T}$ based on direct ones through transitive reasoning. We borrow the notation from description logic to denote the subsumption relationship as $e_1 \sqsubseteq e_2$, meaning that $e_1$ is a sub-class of $e_2$. Under the closed-world assumption, we consider an edge $(e_1, e_2)$ as a *negative sample* if $(e_1, e_2) \notin \mathcal{E} \cup \mathcal{T}$. Particularly, a *hard negative* is identified when $e_1$ and $e_2$ are also siblings that share the same parent.

Explicit hierarchies can often be derived from structured data sources such as taxonomies, ontologies, and knowledge graphs. A taxonomy and the Terminology Box (TBox) of an ontology intrinsically define subsumptions, whereas in knowledge graphs, hierarchical relationships are defined in a more customised manner. For instance, in WordNet, the *hypernym* relationship corresponds to subsumption.

# 3 Hierarchy Transformer Encoder

We intend to propose a general and effective strategy to re-train transformer encoder-based LMs as Hierarchy Transformer encoders (HiTs). To deal with arbitrary input lengths of entity names, we employ an architecture similar to sentence transformers [21], incorporating a mean pooling layer over token embeddings to produce sentence embeddings for entities. Note that some of the sentence transformer models have a normalisation layer after pooling; we exclude this layer because its presence will constrain the embeddings' Euclidean norms to one, thus hindering hierarchical organisation of entity embeddings. It is also worth mentioning that these changes do not add in learnable parameters besides the ones already in LMs, thus retaining the original architectures of LMs as encoders. As aforementioned, the output embedding space of these LMs is typically a

---

[5]E.g., the Poincaré ball model, the Poincaré half plane model, and the hyperboloid model.
[6]An isometry is a bijective distance-preserving transformation.

$d$-dimensional hyper-cube because of the $\tanh$ activation function in the last layer. Thus, we can construct a Poincaré ball of radius $\sqrt{d}$ (or equivalently, curvature value $c = \frac{1}{d}$) whose boundary circumscribes the hyper-cube.[7] Unlike previous hyperbolic embedding methods that utilise the entire hyperbolic space and often require a projection layer to manage out-of-manifold embeddings, our method ensures that embeddings are contained within a specific subset of this manifold. Empirical evidence supports that this subset sufficiently accommodates entities in high-dimensional space (see Section 4.5). Based on this curvature-adapted manifold, we propose the following two losses for hierarchy re-training.

**Hyperbolic Clustering Loss** This loss aims at clustering related entities and distancing unrelated ones in the Poincaré ball. We formulate it in the form of triplet loss because related entities are not equivalent but their semantic distances should be smaller than those between unrelated entities. Formally, the loss is defined as:

$$\mathcal{L}_{cluster} = \sum_{(e, e^+, e^-) \in \mathcal{D}} \max(d_c(\mathbf{e}, \mathbf{e}^+) - d_c(\mathbf{e}, \mathbf{e}^-) + \alpha, 0) \tag{3}$$

Here, inputs are presented in the form of triplet $(e, e^+, e^-)$, where $e^+$ is a parent entity of $e$, and $e^-$ is a negative parent entity of $e$; $\mathcal{D}$ denotes the set of these triplets; $d_c(\cdot, \cdot)$ refers to the hyperbolic distance function defined in Equation (1), and $\alpha$ is the hyperbolic distance margin. The bold letters denote the embeddings of the corresponding entities.

**Hyperbolic Centripetal Loss** This loss ensures parent entities are positioned closer to the Poincaré ball's origin than their child counterparts, reflecting the natural expansion of hierarchies from the origin to the boundary of the manifold. The term *"centripetal"* is used to imply that the manifold's origin represents an **imaginary root entity** for everything. Formally, the hyperbolic centripetal loss is defined as:

$$\mathcal{L}_{centri} = \sum_{(e, e^+, e^-) \in \mathcal{D}} \max(\|\mathbf{e}^+\|_c - \|\mathbf{e}\|_c + \beta, 0) \tag{4}$$

Again, inputs are the triplets in $\mathcal{D}$, but only the child and parent entities (the positive subsumptions) are used to calculate the loss; $\|\cdot\|_c := d_c(\cdot, \mathbf{0})$ refers to the hyperbolic norm; $\beta$ is the hyperbolic norm margin.

The overall hierarchy re-training loss, denoted as $\mathcal{L}_{\text{HIT}}$, is the linear combination of these two hyperbolic losses, defined as:

$$\mathcal{L}_{\text{HIT}} = \mathcal{L}_{cluster} + \mathcal{L}_{centri} \tag{5}$$

In Figure 2, we demonstrate the impact of $\mathcal{L}_{\text{HIT}}$ on entity embeddings. The entity *"e-device"*, being most general, is nearest to the origin. Sibling entities, such as *"phone"* and *"computer"*, *"laptop"* and *"pc"*, are closer to their common parent than to each other, illustrating the effect of re-training to cluster related entities while maintain hierarchical relationships.

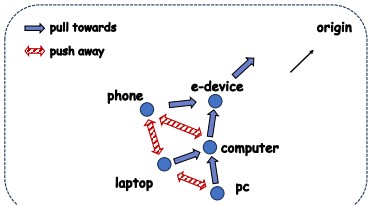

As the HIT model functions as an encoder, it does not inherently support direct predictions of subsumption relationships. To address this, we devise a probing function that leverages the hierarchy re-trained entity embeddings. This function aims to predict the subsumption relationship for any given pair of entities $(e_1, e_2)$, incorporating both the clustering and centripetal heuristics:

Figure 2: Illustration of the impact of $\mathcal{L}_{\text{HIT}}$ during training. In Euclidean space, it seems contradictory that both *"phone"* and *"computer"* are pulled towards *"e-device"* but are also pushed away from each other. However, in principle this is not a problem in hyperbolic space, where distances increase exponentially relative to Euclidean distances as one moves from the origin to the boundary of the manifold.

$$s(e_1 \sqsubseteq e_2) = -(d_c(\mathbf{e}_1, \mathbf{e}_2) + \lambda(\|\mathbf{e}_2\|_c - \|\mathbf{e}_1\|_c)) \tag{6}$$

Here, $\lambda > 0$ represents a weighting factor applied to the centripetal heuristic component. The subsumption score is structured to increase as the hyperbolic distance between $\mathbf{e}_1$ and $\mathbf{e}_2$ decreases,

---

[7]We have also considered scaling down each dimension of the LM embeddings by $\sqrt{d}$ to confine them within a unit Poincaré ball, but we found that losses are harder to converge in this construction.

Table 1: Statistics of WordNet (Noun), Schema.org, and FoodOn, including the numbers of entities (**#Entity**), direct subsumptions (**#DirectSub**), indirect subsumptions (**#IndirectSub**), and the dataset splittings (**#Dataset**) for Multi-hop Inference and Mixed-hop Prediction tasks. Note that the numbers in **#Dataset** are counts of entity pairs rather than entity triplets.

| Source | #Entity | #DirectSub | #IndirectSub | #Dataset (Train/Val/Test) |
|---|---|---|---|---|
| WordNet | 74,401 | 75,850 | 587,658 | multi: 834K/323K/323K
mixed: 751K/365K/365K |
| Schema.org | 903 | 950 | 1,978 | mixed: -/15K/15K |
| FoodOn | 30,963 | 36,486 | 438,266 | mixed: 361K/261K/261K |
| DOID | 11,157 | 11,180 | 45,383 | mixed: 111K/31K/31K |

and/or as the relative difference in their hyperbolic norms ($\|\mathbf{e}_1\|_c - \|\mathbf{e}_2\|_c$) increases. Essentially, for the model to predict $e_1 \sqsubseteq e_2$, it is expected that $\mathbf{e}_1$ and $\mathbf{e}_2$ are relatively closer in the Poincaré ball, with $\mathbf{e}_1$ positioned further from the manifold's origin compared to $\mathbf{e}_2$. The value for $\lambda$ and the overall scoring threshold are to be ascertained through hyperparameter tuning on the validation set.

## 4 Evaluation

### 4.1 Task Definition

**Multi-hop Inference** This task, following the setting in [13], aims to evaluate the model's ability in deducing indirect, multi-hop subsumptions $\mathcal{T}$ from direct, one-hop subsumptions $\mathcal{E}$, so as to simulate transitive inference. We split $\mathcal{T}$ for validation and testing, denoted as $\mathcal{T}_{val}$ and $\mathcal{T}_{test}$, respectively. For each positive subsumption $(e, e^+)$ involved, we sampled 10 negative parents $e^-$ for $e$, leading to 10 training triplets[8]. Following the criteria in Section 2.3, $(e, e^-)$ is a valid negative if $(e, e^-) \notin \mathcal{E} \cup \mathcal{T}$. We further split the task into two settings: one with **random negatives** and another with **hard negatives**, the latter mainly comprising sibling entities. Since not every entity has enough siblings, we supplemented with random negatives that have been sampled to maintain a consistent positive-to-negative ratio of $1 : 10$.

**Mixed-hop Prediction** This task aims to evaluate the model's capability in determining the existence of subsumption relationships between arbitrary entity pairs, where the entities are not necessarily seen during training. We propose a challenging setting where models are trained on incomplete direct subsumptions and examined on a mix of hold-out, unseen direct and indirect (mixed-hop) subsumptions. We split $\mathcal{E}$ into training, validation, and test sets, denoted as $\mathcal{E}_{train}$, $\mathcal{E}_{val}$, and $\mathcal{E}_{test}$, respectively. The final training, validataion, and test sets for this task are $\mathcal{E}_{train}$, $\mathcal{E}_{val} \cup \mathcal{T}_{val}$, and $\mathcal{E}_{test} \cup \mathcal{T}_{test}$, respectively, where $\mathcal{T}_{val}$ and $\mathcal{T}_{test}$ are re-used from the previous task. Again, each positive subsumption in these sets is paired with 10 negative samples, either randomly chosen or from sibling entities. Furthermore, an important factor that reflects the model's generalisability is to examine the **transfer ability across hierarchies**. To this end, we extend the mixed-hop prediction task with a transfer setting where models trained on asserted training edges of one hierarchy are tested on arbitrary entity pairs of another.

**Evaluation Metrics** For both Multi-hop Inference and Mixed-hop Prediction tasks, we utilise Precision, Recall, and F1 score (abbreviated as F-score in latter discussion) as our primary metrics of evaluation. We have opted not to include Accuracy, as preliminary testing indicated a potential bias in this metric, with a misleadingly high score resulting from the much larger volume of negative compared to positive samples. It is important to note that, although the training phase uses entity triplets, the evaluation only involves entity pairs.

### 4.2 Dataset Construction

We constructed the primary dataset from the noun hierarchy of WordNet [14] due to its comprehensive and structured representation of linguistic hierarchies. To assess the transferability and robustness

---

[8]10 training triplets are constructed from 11 entity pairs.

across different domains, we additionally constructed datasets from ontologies that represent varied semantic granularities and domains, namely Schema.org [16], Food Ontology (FoodOn) [17], and Disease Ontology (DOID) [18]. We retrieved WordNet from `NLTK` [25] and adopted pre-processing steps similar to [12], utilising the *hypernym* relations between noun *synsets* to construct the hierarchy. For Schema.org, FoodOn, and DOID, our pre-processing paralleled that in [6], transforming these ontologies into hierarchies of named entities (details in Appendix A). To accommodate the textual input requirements of LMs, we constructed an entity lexicon using the *name* attribute in WordNet and the `rdfs:label` property in the ontologies.[9]

On WordNet (Noun), FoodOn, and DOID, we adopt a consistent splitting ratio for the validation and testing sets. Specifically, we allocate two separate $5\%$ portions of the indirect subsumptions $\mathcal{T}$ to form $\mathcal{T}_{val}$ and $\mathcal{T}_{test}$, respectively. Similarly, two distinct $5\%$ portions of the direct subsumptions $\mathcal{E}$ are used as $\mathcal{E}_{val}$ and $\mathcal{E}_{test}$. As Schema.org is significantly smaller than the other hierarchies and only used for transfer evaluation, we split its entire $\mathcal{E}$ and $\mathcal{T}$ sets into halves for validation and testing, respectively. Table 1 presents the extracted hierarchies' statistics and the resulting datasets for the Multi-hop Inference and Mixed-hop Prediction tasks.

In addition to our main evaluation, we constructed a dataset from the widely-recognised biomedical ontology SNOMED CT [15] and conducted futher evaluation. The relevant details are presented in Appendix D.

### 4.3 Baselines

**Naive Prior**  We first introduce a naive baseline (NaivePrior) that utilises the prior probability of positive subsumptions in the training set for prediction. Given that each positive sample is paired with 10 negatives, the prior probability of a positive prediction stands at $\frac{1}{11}$. Consequently, Precision, Recall, and F-score on the test set are all $\frac{1}{11}$.

**Pre-trained LMs**  We consider pre-trained LMs as baselines to illustrate their limitations in capturing hierarchical structure semantics. As outlined in Section 3, our focus is on LMs based on the sentence transformer architecture [21]. Since these LMs are optimised for cosine similarities between sentences, we devise the following probe for evaluation: for each entity pair $(e_1, e_2)$, we compute the cosine similarity between the masked reference sentence "$e_1$ *is a* $\langle mask \rangle$." and the sample sentence "$e_1$ *is a* $e_2$.". These similarity scores serve as the subsumption scores, with thresholds identified via grid search on the validation set. Note that although these LMs originate from masked language models, they cannot be easily probed via mask filling logits or perplexities as in [26] and [27] because their mask filling layers are not preserved in the released versions. We select three top-performing pre-trained LMs from the sentence transformer library of different sizes, including all-MiniLM-L6-v2 (22.7M), all-MiniLM-L12-v2 (33.4M), and all-mpnet-base-v2 (109M).

**Fine-tuned LMs**  Fine-tuned LMs are used as baselines to demonstrate that despite the efficacy in various tasks, standard fine-tuning struggles to address this specific challenge. Following the BERTSubs approach outlined in [8], we employ pre-trained LMs with an added linear layer for binary classification, and optimising on the Softmax loss. [8] have shown that this method outperforms various structure-based embeddings such as TransE [28] and DistMult [29], and also surpasses OWL2Vec* [30], which integrates both structural and textual embeddings, in subsumption prediction.

**Hyperbolic Baselines**  Previous static hyperbolic embedding models are typically evaluated using the Multi-hop Inference task. In our study, we select the Poincaré Embedding (PoincaréEmbed) [12] and the Hyperbolic Entailment Cone (HyperbolicCone) [13] as baselines on this task. However, their lack of inductive prediction capabilities prevents their evaluation on the Mixed-hop Prediction task and its transfer setting. Additionally, we include the hyperbolic GloVe embedding (PoincaréGloVe) as a baseline in our transfer evaluation. We select the best-performing PoincaréGloVe ($50 \times 2$D with an initial trick) pre-trained on a 1.4B token English Wikipedia dump. While PoincaréGloVe supports inductive prediction, its effectiveness is limited by word-level tokenisation, rendering it less effective at handling unknown words. To address this, we employ NaivePrior as a fallback method for entities that involve unknown words and cannot be predicted by PoincaréGloVe.

More details of our code implementation and experiment settings are presented in Appendix B.

---

[9]We selected the first name (in English) if multiple names for one entity were available.

Table 2: Multi-hop Inference and Mixed-hop Prediction test results on WordNet.

| Model | Random Negatives | | | Hard Negatives | | |
| --- | --- | --- | --- | --- | --- | --- |
| | Precision | Recall | F-score | Precision | Recall | F-score |
| NaivePrior | 0.091 | 0.091 | 0.091 | 0.091 | 0.091 | 0.091 |
| **Multi-hop Inference (WordNet)** | | | | | | |
| PoincaréEmbed | 0.862 | 0.866 | 0.864 | 0.797 | 0.867 | 0.830 |
| HyperbolicCone | 0.817 | 0.996 | 0.898 | 0.243 | 0.902 | 0.383 |
| all-MiniLM-L6-v2 | 0.160 | 0.442 | 0.235 | 0.132 | 0.507 | 0.209 |
| + fine-tune | 0.800 | 0.513 | 0.625 | 0.764 | 0.597 | 0.670 |
| + HIT | 0.864 | 0.879 | 0.871 | 0.905 | 0.908 | 0.907 |
| all-MiniLM-L12-v2 | 0.127 | 0.585 | 0.209 | 0.108 | 0.740 | 0.188 |
| + fine-tune | 0.811 | 0.515 | 0.630 | 0.819 | 0.530 | 0.643 |
| + HIT | 0.880 | 0.927 | 0.903 | 0.910 | 0.906 | 0.908 |
| all-mpnet-base-v2 | 0.281 | 0.428 | 0.339 | 0.183 | 0.359 | 0.242 |
| + fine-tune | 0.796 | 0.501 | 0.615 | 0.758 | 0.628 | 0.687 |
| + HIT | 0.897 | 0.936 | 0.916 | 0.886 | 0.912 | 0.899 |
| **Mixed-hop Prediction (WordNet)** | | | | | | |
| all-MiniLM-L6-v2 | 0.160 | 0.438 | 0.235 | 0.131 | 0.504 | 0.208 |
| + fine-tune | 0.747 | 0.575 | 0.650 | 0.769 | 0.578 | 0.660 |
| + HIT | 0.835 | 0.877 | 0.856 | 0.882 | 0.843 | 0.862 |
| all-MiniLM-L12-v2 | 0.127 | 0.583 | 0.209 | 0.111 | 0.625 | 0.188 |
| + fine-tune | 0.794 | 0.517 | 0.627 | 0.859 | 0.515 | 0.644 |
| + HIT | 0.875 | 0.895 | 0.885 | 0.886 | 0.857 | 0.871 |
| all-mpnet-base-v2 | 0.287 | 0.439 | 0.347 | 0.197 | 0.344 | 0.250 |
| + fine-tune | 0.828 | 0.536 | 0.651 | 0.723 | 0.622 | 0.669 |
| + HIT | 0.892 | 0.910 | 0.900 | 0.869 | 0.858 | 0.863 |

## 4.4 Results

The effectiveness of our hierarchy re-training approach is evident from the results of both the Multi-hop Inference and Mixed-hop Prediction tasks on WordNet (see Table 2), as well as the Transfer Mixed-hop Prediction task on Schema.org, FoodOn, and DOID for pre-trained LMs and models trained on WordNet (see Table 3). In the following, we present several pivotal findings based on these results.

**Performance of HITs**  The HIT models, re-trained from LMs of various sizes, consistently outperform their pre-trained and standard fine-tuned counterparts across all evaluation tasks. In the Multi-hop Inference task, HITs exhibit exceptional performance with F-scores ranging from $0.871$ to $0.916$. This indicates a strong capability in generalising from asserted to transitively inferred entity subsumptions. In the Mixed-hop Prediction task, F-scores ranging from $0.856$ to $0.900$ highlight the effectiveness of HITs in generalising from asserted to arbitrary entity subsumptions. For the Transfer Mixed-hop Prediction tasks, we selected all-MiniLM-L12-v2 as the pre-trained model because all-MiniLM-L12-v2+HIT attains comparable performance to all-mpnet-base-v2+HIT while it is more computationally efficient owing to a smaller parameter size. Notably, all-MiniLM-L12-v2+HIT performs better than pre-trained and fine-tuned all-MiniLM-L12-v2 on these transfer tasks by at least $0.150$ and $0.101$ in F-scores, respectively.

**Limited Hierarchical Knowledge in Pre-trained LMs**  For the tasks on WordNet, all-mpnet-base-v2 achieves the highest F-scores among all the pre-trained models, yet these scores (e.g., $0.347$ and $0.250$ on the Mixed-hop Prediction task with random negatives and hard negatives, respectively) are considerably lower compared to their fine-tuned (lagging by $0.304$ and $0.419$) and hierarchy re-trained (lagging by $0.553$ and $0.613$) counterparts. This disparity confirms findings from LM probing studies such as those by [5] and [6], demonstrating the limited hierarchical knowledge in pre-trained LMs.

Table 3: Transfer Mixed-hop Prediction test results on Schema.org, FoodOn, and DOID.

| Model | Random Negatives | | | Hard Negatives | | |
|---|---|---|---|---|---|---|
| | Precision | Recall | F-score | Precision | Recall | F-score |
| NaivePrior | 0.091 | 0.091 | 0.091 | 0.091 | 0.091 | 0.091 |
| **Transfer Mixed-hop Prediction (WordNet → Schema.org)** | | | | | | |
| PoincaréGloVe | 0.485 | 0.403 | 0.441 | 0.436 | 0.415 | 0.425 |
| all-MiniLM-L12-v2 | 0.312 | 0.524 | 0.391 | 0.248 | 0.494 | 0.330 |
| + fine-tune | 0.391 | 0.433 | 0.411 | 0.597 | 0.248 | 0.351 |
| + HıT | 0.503 | 0.613 | 0.553 | 0.408 | 0.583 | 0.480 |
| **Transfer Mixed-hop Prediction (WordNet → FoodOn)** | | | | | | |
| PoincaréGloVe | 0.192 | 0.224 | 0.207 | 0.189 | 0.200 | 0.195 |
| all-MiniLM-L12-v2 | 0.135 | 0.656 | 0.224 | 0.099 | 0.833 | 0.176 |
| + fine-tune | 0.436 | 0.382 | 0.407 | 0.690 | 0.177 | 0.282 |
| + HıT | 0.690 | 0.463 | 0.554 | 0.741 | 0.385 | 0.507 |
| **Transfer Mixed-hop Prediction (WordNet → DOID)** | | | | | | |
| PoincaréGloVe | 0.265 | 0.314 | 0.287 | 0.283 | 0.318 | 0.299 |
| all-MiniLM-L12-v2 | 0.342 | 0.451 | 0.389 | 0.159 | 0.455 | 0.235 |
| + fine-tune | 0.585 | 0.621 | 0.603 | 0.868 | 0.179 | 0.297 |
| + HıT | 0.696 | 0.711 | 0.704 | 0.810 | 0.435 | 0.566 |

**Limited Generalisation in Fine-tuned LMs**  The research by [8] illustrates that fine-tuned LMs perform well on single-hop subsumptions. Our observations concur, showing that fine-tuned LMs achieve comparable performance as HıTs when assessed on just single-hop test samples. However, their effectiveness wanes when applied to arbitrary entity subsumptions. For the tasks on WordNet, fine-tuned LMs underperform HıTs by $0.194$ to $0.301$ in F-scores. In the transfer task from WordNet to Schema.org, the fine-tuned all-MiniLM-L12-v2 model only marginally outperforms its initial state, with an increase of around $0.02$ in F-scores across both negative settings.

**Performance of Hyperbolic Baselines**  The Multi-hop Inference task with random negatives follows the evaluation in [13]. In this setup, both PoincaréEmbed and HyperbolicCone significantly outperform the pre-trained and standard fine-tuned LMs, and perform comparably to the HıT models. However, HyperbolicCone exhibits substantially worse performance in the hard negative setting; its low precision and high recall suggest difficulties in differentiating sibling entities that are closely positioned in the embedding space. In the transfer evaluation, PoincaréGloVe shows improved performance over pre-trained and standard fine-tuned models on Schema.org. However, it does not demonstrate a similar advantage on FoodOn and DOID, primarily due to its limited vocabulary, which allows it to predict almost all test samples on Schema.org but substantially fewer on the others.

**Comparison of Random and Hard Negatives**  For the tasks on WordNet, hard negative settings present greater challenges compared to random negative settings for all pre-trained LMs. This increased difficulty, however, is not as pronounced in fine-tuned LMs and HıTs. A plausible explanation is that while hard negatives pose challenges, they simultaneously act as high-quality adversarial examples, potentially leading to more robust training outcomes. In the Transfer Mixed Prediction task, hard negative settings are generally more challenging than random negative settings. For instance, in the WordNet-to-DOID transfer task, both fine-tuned and hierarchy re-trained all-MiniLM-L12-v2 models exhibit significantly higher F-scores in the random negative setting, with differences of $0.306$ and $0.138$ respectively, compared to the hard negative setting.

**Case Analysis on WordNet-to-DOID Transfer**  In the WordNet-to-DOID transfer task, the disparity in the *"disease"* category is notable: WordNet contains only $605$ entities that are descendants of *"disease"*, compared to over $10K$ in DOID. Despite this significant difference, HıT models effectively transfer knowledge, achieving F-scores of $0.704$ (random negatives) and $0.566$ (hard negatives).

More discussion on loss functions and an ablation study of loss margins are presented in Appendix C.

## 4.5 Analysis of HIT Embeddings

**Distribution**  Figure 3 illustrates how WordNet entity embeddings generated by all-MiniLM-L12-v2+HIT distribute w.r.t. their hyperbolic norms.

These norms effectively capture the natural expansion of the hierarchical structure, evidenced by an exponential rise in the number of child entities. A notable observation is the sharp decline in the number of entities when hyperbolic norms exceed 23, suggesting that few entities reside at these higher levels. Additionally, the range of entity hyperbolic norms, approximately from 8 to 24, indicates that a relatively small region of the high-dimensional manifold suffices to accommodate all entities in WordNet.

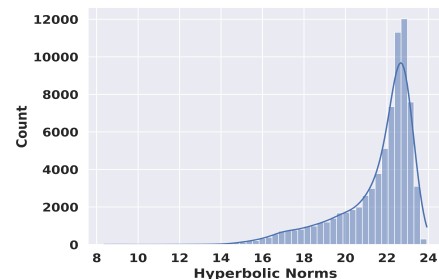

Figure 3: Distribution of WordNet entity embeddings generated by HIT w.r.t. their hyperbolic norms.

**Correlation**  In Table 4, we compare the Pearson correlation coefficients across different hyperbolic models to measure the linear relationship between entities' hyperbolic norms and their depths in WordNet. Our analysis shows that all hyperbolic models lead to a positive correlation between norms and depths, as expected. However, HIT demonstrates a stronger correlation than both PoincaréEmbed and HyperbolicCone.

Table 4: Statistical correlations between WordNet entities' depths and their hyperbolic norms across different hyperbolic models.

| HIT | PoincaréEmbed | HyperbolicCone |
|---|---|---|
| 0.346 | 0.130 | 0.245 |

Table 5: Hyperbolic distances between the embeddings of selected entities (*"computer"*, *"pc"*, *"fruit"*, *"berry"*), along with their individual hyperbolic norms (**h-norm**) and depths in WordNet.

**Case Study**  In Table 5, we showcase the effectiveness of HIT using selected entities: *"computer"*, *"pc"*[10], *"fruit"*, and *"berry"*. The table presents the hyperbolic distances between these entities' embeddings, their individual hyperbolic norms, and their depths[11] in the WordNet hierarchy. We can observe that: *(i)* closely related entities, such as *"fruit"* and *"berry"*, are signifi-

|  | computer | pc | fruit | berry |
|---|---|---|---|---|
| computer | 0.0 | 5.9 | 22.5 | 24.9 |
| pc | 5.9 | 0.0 | 25.2 | 27.2 |
| fruit | 22.5 | 25.2 | 0.0 | 6.72 |
| berry | 24.9 | 27.2 | 6.72 | 0.0 |
| **h-norm** | 17.5 | 19.1 | 15.3 | 16.6 |
| **depth** | 9 | 11 | 9 | 10 |

cantly nearer to each other compared to more distant pairs; *(ii)* more specific entities like *"pc"* and *"berry"* are positioned further from the origin of the manifold than their ancestor entities; *(iii)* the disparity in hyperbolic norms between *"pc"* and *"computer"* is greater compared to that between *"fruit"* and *"berry"*, reflecting the hierarchical depth where *"pc"* is a grandchild of *"computer"*, while *"berry"* is a direct child of *"fruit"*.

## 5  Related Work

Prompt-based probing is widely used for extracting knowledge from LMs. Studies like [5] utilised cloze-style prompts for hypernym detection, while [6] approached subsumption prediction similar to Natural Language Inference. [7] examined if LMs, when correctly predicting *"A is a B"* and *"B is a C"*, can consistently infer the transitive relationship *"A is a C"*. These studies collectively highlight the limited capacity of pre-trained LMs in understanding hierarchical structures. Other research efforts, such as those by [10] and [11], have aimed to incorporate structural semantics into LMs for entity encoding. However, these largely focus on entity equivalence or similarity, with less emphasis on hierarchical organisation.

Regarding hyperbolic embeddings, methods like the Poincaré embedding [12] and the hyperbolic entailment cone [13] have effectively represented hierarchical structures. Despite their efficacy, these techniques are inherently static, constrained by a fixed vocabulary of entities, and do not support

---

[10]The full name *"personal computer"* is used for embedding.

[11]Depth of an entity is the minimum number of hops to the root node. For hierarchies that do not have a root node, we set up an imaginary root node when calculating the depth.

inductive predictions about unseen data. Further explorations include learning word embeddings in hyperbolic space [31, 32]. These methods, however, are limited to word-level tokenisation and yield non-contextual word representations. These shortcomings can be mitigated by integrating hyperbolic embeddings with transformer-based LMs. [33] has explored this direction, applying learnable layers to project LM embeddings into hyperbolic space for syntax parsing and sentiment analysis. Our approach diverges from theirs by focusing on training LMs as general hierarchy encoders without the need for additional learnable parameters.

# 6    Conclusion

This paper tackles the challenge of enabling language models to interpret and encode hierarchies. We devise the hierarchy re-training approach that involves a joint optimisation on both the hyperbolic clustering and hyperbolic centripetal losses, aiming to cluster and organise entities according to their hierarchical relationships. The resulting HIT models demonstrate proficiency in simulating transitive inference and predicting subsumptions within and across hierarchies. Additionally, our analysis of HIT embeddings highlights their geometric interpretability, further validating the effectiveness of our approach.

# 7    Limitations and Future Work

This work does not address the potential loss of pre-trained language understanding resulted from hierarchy re-training. Also, the issue of entity naming ambiguity inherent in the dataset sources is not tackled, which could introduce noise into the training process.

For future work, several promising directions can be pursued: *(i)* investigating methods to measure and mitigate catastrophic forgetting, *(ii)* training a HIT model across multiple hierarchies, either for general or domain-specific applications, *(iii)* extending HIT to accommodate multiple hierarchical relationships within a single model, and *(iv)* developing hierarchy-based semantic search that contrasts with traditional similarity-based approaches.

## Acknowledgments and Disclosure of Funding

This work was supported by Samsung Research UK (SRUK), and EPSRC projects OASIS (EP/S032347/1), UK FIRES (EP/S019111/1), and ConCur (EP/V050869/1). Special thanks to Zifeng Ding for his valuable feedback during the rebuttal process.

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

# A Hierarchy Construction from Ontologies

The Terminology Box (TBox) in an OWL (Web Ontology Language)[12] ontology defines entity relationships using subsumption axioms ($C \sqsubseteq D$) and equivalence axioms ($C \equiv D$), where $C$ and $D$ are **atomic** or **complex** concept expressions defined in the description logic $\mathcal{SROIQ}$. In this work, we used atomic concepts as nodes of the hierarchy. We employed an ontology reasoner to deduce *direct*[13] subsumptions and included these as edges in the hierarchy. This approach fully considers both subsumption and equivalence axioms for all potential edges. Considering an atomic concept Beef, which is defined by Beef $\equiv$ Meat $\sqcap$ $\exists$derivesFrom.Cattle, as an example, the reasoning will lead to an edge between Beef and Meat. Without reasoning, Beef will be misplaced under the root node, if no other subsumptions about Beef are asserted in the ontology. Tools like Protégé[14] also demonstrate similar considerations in presenting ontology concept taxonomies.

It is also important to note the variability in naming schemes across different ontologies, which sometimes necessitates pre-processing of entity names. [6] provided detailed pre-processing steps for Schema.org, FoodOn, and DOID. In this study, we used the pre-processed versions of FoodOn and DOID and applied the same pre-processing methodology to the latest version of Schema.org.

# B Experiment Settings

The code implementation of this work primarily depends on `DeepOnto` [34] for processing hierarchies and constructing datasets, `Geoopt` [35] for Poincaré ball, `Sentence-Transformers` [21] and `Huggingface-Transformers` [36] for training and evaluation of LMs. All our experiments were conducted on a single Quadro RTX 8000 GPU.

In the hierarchy re-training of our HⁱT models, we configured the hyperbolic clustering loss margin ($\alpha$ in Equation 3) at $5.0$ and the hyperbolic centripetal loss margin ($\beta$ in Equation 4) at $0.1$. An exception was made for all-mpnet-base-v2 with hard negatives, where $\alpha$ was adjusted to $3.0$, based on validation. The models were trained for 20 epochs, with a training batch size of 256, 500 warm-up steps and an initial learning rate of $10^{-5}$, using the AdamW optimiser [37]. Model selection was conducted after each epoch, guided by performance on the validation set.

For standard fine-tuning, we largely adhered to the default settings of the Huggingface Trainer[15], but maintained the same training batch size as used in the hierarchy re-training. Notably, our preliminary testing revealed that fine-tuning is more prone to overfitting. Consequently, we adopted a more frequent model selection interval, performing this assessment every 500 training steps rather than epoch-wise.

For our hyperbolic baselines, we trained PoincaréEmbed with an embedding dimension of 200 for 200 epochs, a training batch size of 256, 10 warm-up epochs, and a constant learning rate of 0.01, using the Riemannian Adam optimiser [38]. According to [13], HyperbolicCone benefits from a robust initialisation such as the one provided by a pre-trained PoincaréEmbed. Consequently, we utilised the same hyperparameters to train HyperbolicCone except that we initialised it with the weights from our pre-trained PoincaréEmbed. As outlined in the main paper, we selected the optimal pre-trained PoincaréGloVe model reported by [32] (50×2D with an initial trick) as a baseline for our transfer evaluation.

# C Further Discussion on $\mathcal{L}_{\text{HⁱT}}$

**Loss Variants**  As discussed in the main paper, we opted for the triplet contrastive loss format for hierarchy re-training because hierarchically related entities (e.g., subsumptions) should be closer together, yet not equivalent. We tested other forms of loss, including standard contrastive loss, which minimises absolute hyperbolic distances between related entities, and softmax contrastive loss, which

---

[12]https://www.w3.org/OWL/

[13]Refer to the definition of DirectSubClassOf at https://owlcs.github.io/owlapi/apidocs_4/org/semanticweb/owlapi/reasoner/OWLReasoner.html.

[14]https://protege.stanford.edu/

[15]https://huggingface.co/docs/transformers/main_classes/trainer

was adopted in training PoincaréEmbed [12]. Our trial experiments demonstrated that the triplet form converges more efficiently and effectively compared to these alternatives.

**Loss Margins** We provide an ablation study on loss margins $\alpha$ and $\beta$ defined in Equation (3) and Equation (4), respectively.

Table 6: Ablation results (F-score) of allMiniLM-L12-v2+HıT on WordNet's Mixed-hop Prediction.

| $\alpha = 5.0, \beta = 0.1$ | $\alpha = 3.0, \beta = 0.1$ | $\alpha = 1.0, \beta = 0.1$ | $\alpha = 5.0, \beta = 0.5$ |
|---|---|---|---|
| 0.885 | 0.865 | 0.867 | 0.899 |

The results from Table 6 indicate that although loss margins impact performance, the HıT model exhibits robustness to their variations. Notably, a higher F-score for $\alpha = 5.0, \beta = 0.5$ is observed, surpassing the results presented in the main paper. Despite this, we chose not to overly optimise $\alpha$ and $\beta$ to avoid overfitting and to maintain the generalisability of our findings.

# D    Results on SNOMED CT

In the main paper, we primarily focused on models trained on the WordNet (Noun). This section broadens our study to encompass models trained on SNOMED CT [15], a structured, comprehensive, and widely-used vocabulary for electronic health records. Using the tool provided by the SNOMED CT team,[16] we converted the latest version of SNOMED CT (released in December 2023) into the OWL ontology format. We then constructed the hierarchy according to the procedure detailed in Appendix A.

For entity name pre-processing in SNOMED CT, we addressed potential information leakage during testing. Typically, an SNOMED CT entity is named in the format of *"⟨entity name⟩ (⟨branch name⟩)"*, e.g., *"virus (organism)"*. The *branch name* denotes the top ancestor and is propagated through all its descendants. To prevent this information from biasing the model, we removed the branch name from each entity.

Table 7: Statistics of SNOMED-CT, including the numbers of entities (**#Entity**), direct subsumptions (**#DirectSub**), indirect subsumptions (**#IndirectSub**), and the dataset splittings (**#Dataset**) for Multi-hop Inference and Mixed-hop Prediction tasks.

| Source | #Entity | #DirectSub | #IndirectSub | #Dataset (Train/Val/Test) |
|---|---|---|---|---|
| SNOMED | 364,352 | 420,193 | 2,775,696 | mixed: 4,160K/1,758K/1,758K |

In Table 7, we present relevant statistics of the SNOMED CT hierarchy and its corresponding dataset, which was constructed using the method outlined in Section 4.2.

Table 8 details the Mixed-hop Prediction results on SNOMED CT, along with the Transfer Mixed-hop Prediction results on Schema.org, FoodOn, and DOID for pre-trained LMs and models trained on SNOMED CT. Building on the demonstrated effectiveness of HıT in simulating transitive inference from the main body of this paper, we present only the results for inductive subsumption prediction. The transfer evaluation incorporates the same set of hierarchies as those used in the evaluation on WordNet (Noun), facilitating a meaningful comparison of model performance across different training hierarchies.

In the Mixed-hop Prediction task on SNOMED CT, all models–pre-trained, fine-tuned, and hierarchy re-trained–outperform their counterparts on WordNet (Noun) in terms of F-scores. This improved performance is likely to be attributed to SNOMED CT's more precise entity naming and better organised concept hierarchy, which reduces both textual and structural ambiguity.

In the transfer results, models trained on SNOMED CT exhibit notably better F-scores on DOID, which aligns with expectations given DOID's focus on diseases and SNOMED CT's comprehensive biomedical scope. Conversely, their performance on Schema.org, a common-sense ontology, is comparatively worse. Notably, the fine-tuned all-MiniLM-L12-v2 performs even worse than its

---

[16]https://github.com/IHTSDO/snomed-owl-toolkit

Table 8: Mixed-hop Prediction test results on SNOMED and Transfer Mixed-hop Prediction results on Schema.org, FoodOn, and DOID.

| Model | Random Negatives | | | Hard Negatives | | |
|---|---|---|---|---|---|---|
| | Precision | Recall | F-score | Precision | Recall | F-score |
| MajorityPrior | 0.091 | 0.091 | 0.091 | 0.091 | 0.091 | 0.091 |
| **Mixed-hop Prediction (SNOMED)** | | | | | | |
| all-MiniLM-L12-v2 | 0.224 | 0.443 | 0.297 | 0.145 | 0.398 | 0.213 |
| + fine-tune | 0.919 | 0.859 | 0.888 | 0.894 | 0.635 | 0.743 |
| + HiT | 0.941 | 0.967 | 0.954 | 0.905 | 0.894 | 0.899 |
| **Transfer Mixed-hop Prediction (SNOMED → Schema.org)** | | | | | | |
| all-MiniLM-L12-v2 | 0.312 | 0.524 | 0.391 | 0.248 | 0.494 | 0.330 |
| + fine-tune | 0.198 | 0.864 | 0.322 | 0.431 | 0.288 | 0.345 |
| + HiT | 0.432 | 0.580 | 0.495 | 0.274 | 0.608 | 0.378 |
| **Transfer Mixed-hop Prediction (SNOMED → FoodOn)** | | | | | | |
| all-MiniLM-L12-v2 | 0.135 | 0.656 | 0.224 | 0.099 | 0.833 | 0.176 |
| + fine-tune | 0.378 | 0.540 | 0.445 | 0.638 | 0.371 | 0.469 |
| + HiT | 0.700 | 0.500 | 0.583 | 0.594 | 0.442 | 0.506 |
| **Transfer Mixed-hop Prediction (SNOMED → DOID)** | | | | | | |
| all-MiniLM-L12-v2 | 0.342 | 0.451 | 0.389 | 0.159 | 0.455 | 0.235 |
| + fine-tune | 0.547 | 0.912 | 0.684 | 0.831 | 0.795 | 0.812 |
| + HiT | 0.836 | 0.864 | 0.850 | 0.739 | 0.748 | 0.744 |

pre-trained version on Schema.org, suggesting an overfitting to biomedical domain knowledge in SNOMED CT. In contrast, HiT demonstrates greater robustness against such overfitting.

