# OpenReview forum: "Language Models as Hierarchy Encoders"
_NeurIPS.cc/2024/Conference — NeurIPS 2024 poster_

### Official Review · Reviewer_DpLC · 2024-07-08

**Soundness:** 3
**Presentation:** 3
**Contribution:** 2
**Rating:** 5
**Confidence:** 3

**Summary:**

This paper proposes a method that leverages a non-Euclidean representation space to better encode hierarchical relationships between entities. Specifically, a hyperbolic embedding space is defined, wherein items closer to the origin are higher-level concepts, and items further from the origin are lower-level concepts; items closer to each other typically have closer semantic relationships, such that it is then straightforward to translate between geometric distances and inheritance relationships between concepts—even when the relationship was not explicitly encoded in the training set.

In a series of experiments, the authors leverage sentence-Transformer models as bases to predict whether two entities have taxonomic relationships in a zero-shot manner. Comparisons are performed between the base sentence-Transformer models, the same models with task-specific fine-tuning, and the proposed hyperbolic encoding method. It is observed that the proposed method outperforms more naive methods.

**Strengths:**

* The proposed embedding space works well as a way to categorize hierarchical relationships between inputs. It also makes it easy to tell what the relationship is between two concepts in latent space, as distances between concepts and distances to the origin both have well-defined meanings in this space.
* The method can be run with modest compute resources, making it accessible and easily scalable as sentence-Transformer models scale and improve.
* The idea is interesting and, to my knowledge, novel.

**Weaknesses:**

1. As the authors acknowledge, it is not clear what other knowledge is lost when moving to this new latent space. While hierarchical relationships between entities are now easy to understand, does the new space also preserve properties of each independent entity? Evaluations on a wider variety of downstream tasks would be helpful for establishing what kinds of semantic information are preserved.
2. The analysis in Table 4 could have been cherry-picked. A more systematic version of the analysis in Table 4 is possible, and would be nice to see: specifically, one could search over a larger subset of WordNet, quantifying whether more deeply nested concepts correlate with higher h-norms, and whether more closely related concepts have significantly lower hyperbolic distances than unrelated but semantically similar concepts.
3. When we classify whether two entities are taxonomically related, we usually care more about whether we can leverage these relationships to improve performance on some other metric. For example, better representational quality could lead to better deductive reasoning. Consider, for example, a counterfactual reasoning task where we say “Birds have fur. Is it therefore true that sparrows have fur?” A model that better captures taxonomic relationships should be better at this task. What I’m getting at is that it would be nice to evaluate on downstream tasks that require entity classification as a key part, rather than directly evaluating entity classification (a rare task). This would demonstrate the utility of this method, and increase interest and impact among a broader audience.

**Questions:**

1. Why sample to a consistent 1:10 ratio of positive:negative examples? Was this decision made empirically?
2. How long did it take to run your proposed method compared to standard fine-tuning? Is there a better trade-off between performance and compute?

**Limitations:**

Yes.

---

> ### Author Rebuttal · Authors · 2024-08-03
>
> Thank you for your review and feedback. We address your comments and questions below:
>
> ------
>
> **Regarding Weakness 1**:
>
> Our current focus in this paper is on enabling transformer encoder-based language models to explicitly encode hierarchies. We recognise the importance of evaluating the preservation of other semantic information. In our future work, we plan to explore how our method impacts semantic properties unrelated to hierarchies, to ensure a balanced representation that retains existing language understanding.
>
> ------
>
> **Regarding Weakness 2**:
>
> We compare the Pearson correlation coefficients across different hyperbolic models to measure the linear relationship between entities' hyperbolic norms and their depths in WordNet. Our analysis shows that all hyperbolic models lead to a positive correlation between norms and depths, as expected. However, our HiT model demonstrates a stronger correlation than both PoincaréEmbed and HyperbolicCone.
>
> |HiT | PoincaréEmbed | HyperbolicCone |
> |:-------------------------:|:-----------------------------:|:----------------------------:|
> | 0.346                   | 0.130                       | 0.245                      |
>
> >Table: Statistical correlations between WordNet entities' depths and their hyperbolic norms in HiT, Poincaré Embed, and HyperbolicCone, respectively.
>
> We will add this analysis in the revised manuscript.
>
> The **Hard Negative** setting for each task is designed to determine if our models can distinguish unrelated but semantically similar concepts (e.g., sibling entities), as you suggested. Our results demonstrate that HiT models consistently outperform other baselines in this setting. The hyperbolic distances between closely related concepts will not be significantly lower than unrelated concepts because our loss functions are optimised for relative differences, but they are sufficiently distinct to enable good predictions.
>
> ------
>
> **Regarding Weakness 3**:
>
> Our current evaluation tasks aim to understand if models can generalise from asserted subsumptions to inferred (transitive) and unseen (inductive) subsumptions, which is a critical capability for completing missing knowledge or enriching new knowledge in taxonomies and hierarchies. We acknowledge the importance of further downstream tasks to demonstrate more real-world utilities. Taking your counterfactual reasoning example, it can be formulated as predicting if “sparrow” is subsumed by “something that has fur”, which can be seen as an existential restriction (a kind of complex concept) in the context of ontology. We will extend our settings to handle complex, non-atomic entities in future work.
>
>
> ------
>
> **Regarding Question 1**:
>
> We followed the evaluation setting in [1] to maintain consistency with existing hyperbolic baselines.
>
> ------
>
> **Regarding Question 2**:
>
> Using our GPU resources (see Appendix), taking all-MiniLM-L6-v2 as the base model, hierarchy re-training takes approximately 65 minutes, while standard fine-tuning takes about 17 minutes. Standard fine-tuning requires only a few epochs for convergence, but its performance is capped and cannot be further improved. Early stopping is a possible optimisation for hierarchy re-training to save time, but it was not considered in this work. We will explore this and other optimisation strategies in future work.
>
> ------
>
> - [1] Ganea, Octavian, Gary Bécigneul, and Thomas Hofmann. "Hyperbolic entailment cones for learning hierarchical embeddings." ICML (2018).

---

> > ### Comment · Reviewer_DpLC · 2024-08-12
> >
> > Thank you for the thorough response. I appreciate the extra analysis in response to Weakness 2, and consider this addressed.
> >
> > The rest of the points would be very helpful in demonstrating more general impact, but I suppose these would be effort-intensive. As-is, the paper feels like it could benefit greatly from more explicitly evaluating generalizability outside of this narrow task setting. I therefore would like to keep my score the same. That said, if the paper is borderline after taking all other reviews into account, consider my vote in favor of accepting.

---

> > > ### Author Response · Authors · 2024-08-13
> > >
> > > Thank you for your thoughtful feedback. We appreciate your acknowledgment of the additional analysis addressing Weakness 2 and your willingness to consider the paper positively.

---

### Official Review · Reviewer_c8m8 · 2024-07-13

**Soundness:** 3
**Presentation:** 3
**Contribution:** 2
**Rating:** 4
**Confidence:** 3

**Summary:**

The paper introduces a method to re-train transformer-based language models as Hierarchy Transformer encoders (HITs), using the properties of hyperbolic space to enhance their ability to encode hierarchical structures in language.

**Strengths:**

1. The utilization of hyperbolic space to encode hierarchical structures in language models is a creative and theoretically sound approach, as hyperbolic space naturally lends itself to representing hierarchies.
2. The paper provides extensive experimental results, showing that HITs consistently outperform traditional pre-trained and fine-tuned models across multiple datasets and tasks.
3. The methodology, experimental setup, and results are clear, making the paper accessible to readers.

**Weaknesses:**

1. The motivation is a bit overstated since it is heavily on the claim that current language models are significantly deficient in encoding hierarchical information. Thus, improvements can certainly be made.
2. The application of hyperbolic spaces in language models, while interesting, isn't entirely quite inspiring as previous works have explored similar ideas. The distinction between this and prior approaches isn't as significant as suggested.
3. The downstream tasks selected by this study have deviated from those for the LLMs.

**Questions:**

See weaknesses.

**Limitations:**

Yes

---

> ### Author Rebuttal · Authors · 2024-08-02
>
> Thank you for your review and feedback. We address your comments and questions below:
>
> ------
>
> **Regarding Weakness 1**:
>
> In Introduction, we acknowledge that hierarchical information has been considered in existing language model studies. Our claim emphasises the lack of **explicit geometric interpretability in hierarchical encoding**. For instance, as highlighted in [1], pre-trained language models often fail to capture the transitivity property inherent in hierarchical relationships. Our experiments demonstrate that standard fine-tuning, despite its strengths, cannot effectively capture this transitivity (and does not have suffcient interpretability) compared to our proposed HiT method. We will revise our contribution statement to make it clearer.
>
> ------
>
> **Regarding Weakness 2**:
>
> The major differences of our work compared to existing related works are discussed in Sections 1 & 5. In particular, we mentioned that:
>
> - Previous works on pre-trained language models considering hierarchies did not focus on explicit geometric interpretability (line 338); while our work seeks to construct a language model-based hierarchy encoder with such interpretability.
> - Previous hyperbolic embeddings did not support inductive prediction (line 343) or did not handle unseen entities effectively (line 344-345); while our models can naturally support inductive prediction on unseen data.
> - [2] investigated adding a downstream hyperbolic layer to pre-trained LMs for classification tasks, whereas our work focuses on explicitly encoding hierarchies without requiring additional trainable parameters (line 346-349).
>
> ------
>
> **Regarding Weakness 3**:
>
> Our evaluation tasks are designed to assess whether **transformer encoder-based language models** can be trained to capture hierarchical structures by their ability to generalise from asserted subsumptions to inferred and unseen subsumptions. Our mixed-hop prediction task, which involves predicting missing subsumptions between arbitrary entities (potentially unseen), is a critical knowledge-intensive task for completing or enriching knowledge in hierarchies and taxonomies.
>
> ------
>
> - [1] Ruixi Lin and Hwee Tou Ng. “Does bert know that the is-a relation is transitive?” ACL (2022).
> - [2] Boli Chen, Yao Fu, Guangwei Xu, Pengjun Xie, Chuanqi Tan, Mosha Chen, and Liping Jing. “Probing bert in hyperbolic spaces.” ICLR (2020).

---

### Official Review · Reviewer_9j2x · 2024-07-13

**Soundness:** 3
**Presentation:** 4
**Contribution:** 3
**Rating:** 6
**Confidence:** 4

**Summary:**

This paper proposes a new way to retrain encoder-based language models into hierarchy encoders. Specifically, they propose to recast the output embedding space onto a Poincaré ball and retraining with the designed loss functions for organizing entities into hierarchy. The experiments on real world datasets like WordNet demonstrates the advantages of the proposed approach.

**Strengths:**

- The paper is well-written, scoped, and nicely organized.
- The topic of addressing hierarchy with language models is interesting.
- The proposed approach is novel and the corresponding experimental results validate the effectiveness of the proposed method.

**Weaknesses:**

- Can you explain if the proposed method can be used to decoder-only models?
- Will frequency of entities in the data impact the hierarchy captured in the representations?

**Questions:**

- see. weakness

**Limitations:**

The authors discuss their limitation in the sec 6.

---

> ### Author Rebuttal · Authors · 2024-08-02
>
> Thank you for your review and feedback. We address your comments and questions below:
>
> -------
>
> **Regarding Weakness 1**:
>
> Our current method is specifically designed for encoder-based language models due to their ability to produce embeddings with **more straightforward semantic meanings**. Decoder-based models, which generate tokens based on previously generated ones, present a **challenge in directly editing their embeddings to assign geometric interpretability**. Retrieving meaningful semantic embeddings from decoder-based LLMs is itself a challenging research question. This is an area we recognise as valuable for future research. Recent works like [1] attempt to enable typical encoder training on decoder-based LLMs by modifying attention mechanisms and specifically masking the next token. We believe our approach **can be adapted to the decoder-based framework when a more systematic way of embedding extraction from decoder-only models is developed**. Despite this, our encoder-based models can still support decoder-based models by providing hierarchy-aware context, which can enhance the generation process.
>
> -------
>
> **Regarding Weakness 2**:
>
> Frequent entities receive more exposure during training, resulting in more detailed and fine-grained hierarchical representations. This can lead to a similar challenge faced in many machine learning works: handling long-tail entities that appear less frequently. However, as our models are re-trained from pre-trained LMs, the distribution of entities in the pre-training data also has an impact. Addressing this aspect is beyond the current scope of our work.
>
> -------
>
> - [1] BehnamGhader, Parishad, Vaibhav Adlakha, Marius Mosbach, Dzmitry Bahdanau, Nicolas Chapados, and Siva Reddy. "LLM2vec: Large language models are secretly powerful text encoders." arXiv (2024).

---

> > ### Comment · Reviewer_9j2x · 2024-08-10
> > **thank you for the rebuttal.**
> >
> > thank you for your response.

---

> ### Author Response · Authors · 2024-08-11
>
> We sincerely appreciate your response and wish our rebuttal has addressed your concerns.

---

### Official Review · Reviewer_fC9f · 2024-07-16

**Soundness:** 3
**Presentation:** 3
**Contribution:** 2
**Rating:** 5
**Confidence:** 4

**Summary:**

This paper presents a novel approach called Hierarchy Transformer encoders (HITs) to retrain transformer-based language models to better encode hierarchical structures in language. The method involves situating the output embedding space of pre-trained language models within a Poincaré ball and training on hyperbolic clustering and centripetal losses. The authors evaluate HITs against pre-trained LMs, fine-tuned LMs, and hyperbolic embedding baselines on tasks including multi-hop inference and mixed-hop prediction. Results show that HITs consistently outperform baselines, demonstrating improved ability to capture hierarchical relationships and generalize across hierarchies.

**Strengths:**

1. The paper introduces a novel method to explicitly encode hierarchies in language models without adding new parameters.

2. The authors conduct extensive experiments across multiple tasks and datasets, and show the effectiveness of their approach.

**Weaknesses:**

1. The proposed method requires predefined hierarchies which may limit its generalization, especially when new entity arrives while no relation is presented for the new entity.

2. Applications of the learnt entity embedding are desired to show the effectiveness of the embedding. For example, how can the embedding be used in downstream tasks.

**Questions:**

Could you show more examples on how hierarchies are encoded through the proposed approach?

**Limitations:**

Yes

---

> ### Author Rebuttal · Authors · 2024-08-02
>
> Thank you for your review and feedback. We address your comments and questions below:
>
> --------
>
> **Regarding Weakness 1**:
>
> Our HiT models **can deal with new entities** and the capability of predicting subsumptions between arbitrary entity pairs is one of the key highlights. Our Mixed-hop prediction task specifically examines a model's ability to make **inductive predictions involving unseen, new entities**. During evaluation, we exclude 10% of the subsumptions, which can include these unseen entities, to test this capability. Additionally, we conduct transfer evaluations across three different hierarchies. In all these settings, the HiT models significantly surpass the baselines, demonstrating their ability to generalise and handle new entities effectively.
>
> --------
>
> **Regarding Weakness 2**:
>
> The primary focus of this work is to explore **explicit hierarchy encoding** within **transformer encoder-based language models**. Our evaluation tasks are tailored for this purpose and align with evaluation in prior works like [1] and [2], which aim to construct **geometrically interpretable** hierarchy embeddings, while extending to real-world scenarios where entities and relationships can be new and unseen. Predicting subsumptions and other hierarchical relationships between arbitrary entities is a crucial task for completing missing knowledge or enriching new knowledge in taxonomies and hierarchies, as well as querying hierarhical contexts from them. We will explore more specific use-cases and downstream tasks of HiTs in future work.
>
> --------
>
> **Regarding Question 1**:
>
> Below shows two example hierarchical paths in WordNet:
>
> ```
> person -> actor -> comedian
> fluid -> liquid -> water
> ```
>
> and their embeddings have the following statistics:
>
> |          |   person |   actor |   comedian |   fluid |   liquid |   water |
> |:---------|---------:|--------:|-----------:|--------:|---------:|--------:|
> | **person**   |      0   |     5.3 |       12   |    15   |     15.5 |    20   |
> | **actor**    |      5.3 |     0   |       10.9 |    18.2 |     18.7 |    22.9 |
> | **comedian** |     12   |    10.9 |        0   |    21.5 |     21.3 |    25.5 |
> | **fluid**    |     15   |    18.2 |       21.5 |     0   |      5.1 |    11.1 |
> | **liquid**   |     15.5 |    18.7 |       21.3 |     5.1 |      0   |     9   |
> | **water**    |     20   |    22.9 |       25.5 |    11.1 |      9   |     0   |
> | *h-norms*  |     13.6 |    15.5 |       19.2 |    17.3 |     17.4 |    19.8 |
> > Table: Hyperbolic distances between the embeddings of entities in above examples, along with their individual hyperbolic norms.
>
> We can see that the hyperbolic norms of entity embeddings generally follow the hierarchical paths and related entities are closer than non-related ones.
>
> --------
>
> - [1] Nickel, Maximillian, and Douwe Kiela. "Poincaré embeddings for learning hierarchical representations." NeurIPS (2017).
> - [2] Ganea, Octavian, Gary Bécigneul, and Thomas Hofmann. "Hyperbolic entailment cones for learning hierarchical embeddings." ICML (2018).

---

> > ### Comment · Reviewer_fC9f · 2024-08-13
> >
> > Thanks for your response. I keep my score unchanged.

---

> > > ### Author Response · Authors · 2024-08-13
> > >
> > > Thank you for your response. We appreciate your feedback and hope that our rebuttal has adequately addressed your concerns.

---

### Decision · Program_Chairs · 2024-09-25

**Decision:**

Accept (poster)

**Comment:**

This paper proposes a novel method that retrains transformer encoders to better encode structures in language. The output embedding space is mapped onto a Poincaré ball, and the model is trained with specially designed objectives. Experiments show that the proposed approach outperforms baselines in tasks requiring the encoding of hierarchical structures.

Most reviews are ambivalent. The novelty and clarity of the work are appreciated by all reviewers. With the authors’ additional clarifications and experiments during the discussion period, most of the reviewers’ concerns have been addressed. However, a major concern that remains is its broad applicability, including other tasks and decoder architectures. Such efforts will help the paper appeal to a broader audience, but perhaps out of the scope. Overall, I recommend that this paper be accepted, and encourage the authors to incorporate the discussion into the revision.

At least one review is discounted due to lack of participation in the discussion period.